# Enhancing Bioactivity through the Transfer of the 2-(Hydroxymethoxy)Vinyl Moiety: Application in the Modification of Tyrosol and Hinokitiol

**DOI:** 10.3390/molecules29143414

**Published:** 2024-07-21

**Authors:** Marcin Molski

**Affiliations:** Department of Quantum Chemistry, Faculty of Chemistry, Adam Mickiewicz University of Poznań, ul. Uniwersytetu Poznańskiego 8, 61-614 Poznań, Poland; mamolski@amu.edu.pl

**Keywords:** radical scavengers, preservatives, formaldehyde releasers, thermodynamic descriptors, chemical activity descriptors, meso-carriers, tyrosol, hinokitiol

## Abstract

Utilizing Density Functional Theory (DFT) calculations at the B3LYP/QZVP level and incorporating the Conductor-like Polarizable Continuum Model (C-PCM) for solvation, the thermodynamic and chemical activity properties of 21-(hydroxymethoxy)henicosadecaenal, identified in cultured freshwater pearls from the mollusk *Hyriopsis cumingii*, have been elucidated. The study demonstrates that this compound releases formaldehyde, a potent antimicrobial agent, through dehydrogenation and deprotonation processes in both hydrophilic and lipophilic environments. Moreover, this polyenal exhibits strong anti-reductant properties, effectively scavenging free radicals. These critical properties classify the pearl-derived ingredient as a natural multi-functional compound, serving as a coloring, antiradical, and antimicrobial agent. The 2-(hydroxymethoxy)vinyl (HMV) moiety responsible for the formaldehyde release can be transferred to other compounds, thereby enhancing their biological activity. For instance, tyrosol (4-(2-hydroxyethyl)phenol) can be modified by substituting the less active 2-hydroxyethyl group with the active HMV one, and hinokitiol (4-isopropylotropolone) can be functionalized by attaching this moiety to the tropolone ring. A new type of meso-carrier, structurally modeled on pearls, with active substances loaded both in the layers and the mineral part, has been proposed.

## 1. Introduction

Multi-functional ingredients of medicines, cosmetics, supplements, and food are the subject of intensive research and present a significant challenge for modern life sciences [1]. Components of these types can reduce the quantity of the product’s base compounds, thereby minimizing chemical interactions that can lead to uncontrolled changes in composition and, consequently, undesirable side effects. There are three main sources of these new multi-functional compounds: (i) natural raw materials of plant, fungi, and animal origin, in which they typically occur as holistic complexes of bioactive substances, (ii) the synthesis of the dominant compound in the complex that “mimics” the activity of the natural raw material, and (iii) the modeling of new compounds not occurring in nature, which have a structure similar to their natural counterparts but exhibit greater bioactivity than the parent compounds. Research in this area primarily focuses on more potent antioxidants (anti-reductants), preservatives, and UV protectors as well as anti-aging, anti-inflammatory, and anti-cancer agents as potential ingredients of cosmetics and food products, medicinal formulations, and treatment-supporting supplements [2,3]. Greater bioactivity results in a lower concentration of bioactive components in the product, consequently reducing its toxicity and allergenicity. Investigations in this field have revealed that the best candidates for multi-functional compounds are conjugated polyenes, which can be viewed as chemical probes (biomarkers) of life signatures [4]. These compounds occur in numerous organisms, including plants, fungi, and animals, and are responsible for their coloring and protection against microorganisms, photoinhibition, UV radiation, and radical damage to DNA, RNA, lipids, cells, and tissues [5,6,7,8]. The class of conjugated polyenes includes such families of compounds as tetraterpenes (carotenoids) [4], unmethylated linear polyenals (polyunsaturated aldehydes) [9,10], and laetiporic acids [4]. Of the aforementioned compounds, polyenals are the least studied class, with a great application potential. They have been identified in parrot feather pigments (6–9 conjugated double bonds in the carbon chain) [9,10], pearls, mollusk shells, and octocorals (6–14 double bonds) [11,12,13,14,15]. Research in this field [15] has led to the discovery of a polyenal 21-(hydroxymethoxy)henicosadecaenal (abbreviated as PE, see Figure 1) in cultured freshwater pearls from the mollusk *Hyriopsis cumingii*.

In addition to its role as a coloring substance responsible for the color of the pearls, its biological functions and chemical reactivity have not been fully elucidated yet. Raman spectroscopy has confirmed the presence of PE in orange and purple-colored pearls, whereas in white pearls, it was undetectable. Studies employing Plasma Emission Spectroscopy have revealed that the coloration of pearls is closely related to the type and concentration of trace metal ions, which occur in the form of organometallic compounds such, as metalloporphyrins. For instance, white pearls lack detectable levels of Ti, V, and Ag (<0.1 [mg kg^−1^]), the presence of these elements can be observed in orange (Ti: 1.46 [mg kg^−1^), V: 1.31 [mg kg^−1^]) and purple (Ti: 8.10 [mg kg^−1^], V: 1.65 [mg kg^−1^]), Ag: 4.20 [mg kg^−1^]) pearls [15].

The chemical structure of the compound in question is similar to parrodienes (psittacofulvins), which have been identified in the colored feathers of parrots and exhibit anti-radical, antimicrobial, and anti-inflammatory activities. These compounds perform important biological functions such as aposematism (warning display) and crypsis (camouflage). Furthermore, bird coloration based on carotenoids and parrodienes plays a role in sexual selection, with brightly colored males being more attractive to females or outcompeting other males in contests. This raises the question of what biological functions the PE component of pearls serves when concealed within the shell and isolated from the external environment? To address this, it is necessary to analyze the morphogenesis of pearls, coupled with the reactivity characteristics of the PE compound responsible for the pearl coloration. Pearls are formed as a natural defense mechanism of the host organism against an irritant, such as a grain of sand, organic materials, tissue damage, or a parasite entering the shell. This mechanism involves coating the irritant with layers of aragonite or a mixture of aragonite and calcite (polymorphs of calcium carbonate, CaCO_3_) and the gluing protein conchiolin, thereby encasing the irritant and protecting the mollusk’s tissue from potential harm. The presence of PE in this mineral–protein structure suggests that the PE plays a significant role in the mollusk’s protective system. To elucidate this role, a key task is to determine the chemical and thermodynamic characteristics of PE, which will provide essential information on its general chemical and physical reactivity and its specific bioactivity. Given that PE has two chemically active moieties—the aldehyde group and the hydroxyl group—the analysis must characterize both moieties. It will be demonstrated that PE, through dehydrogenation and deprotonation processes in hydrophilic and lipophilic environments, releases formaldehyde—a potent antimicrobial agent. Furthermore, it will be shown that PE is a strong anti-reductant capable of scavenging free radicals. These two key properties classify PE as a natural multi-functional compound that acts as a coloring, antiradical, and antimicrobial agent. This serves as a prelude to the second part of the work, in which the most active moiety identified in the first stage of the study will be utilized in the modeling of multi-functional compounds exhibiting both antiradical and antimicrobial properties similar to those of psittacofulvins [5,6,7,8]. As an illustrative case, this study examines the structural modification of tyrosol (4-(2-hydroxyethyl)phenol, abbreviated as T, with TD signifying its derivative), classified as a weak [16] or moderate [17,18] radical scavenger, and hinokitiol (4-isopropyl tropolone, abbreviated as H, with HD signifying its derivative), a potent antimicrobial agent [19,20] and a weak radical scavenger [21]. The proposed modifications involve (i) substituting the inactive 2-hydroxyethyl group in T with the HMV moiety responsible for the release of formaldehyde, and (ii) attaching the HMV group to the tropolone ring in H. They are anticipated to extend (T→TD) and enhance (H→HD) the bioactive properties of the parent compounds by incorporating the antimicrobial HMV moiety (see Figure 2).

In summary, the primary objectives of this research are as follows: (i) to determine the chemical and thermodynamic activity descriptors of PE, T, and H; (ii) to demonstrate that PE functions as both a radical scavenger and a natural formaldehyde releaser; (iii) to enhance the bioactivity of T and H by transferring the HMV moiety (responsible for formaldehyde release) from PE to T and H; (iv) to determine the activity parameters of the designed TD and HD and compare them with the descriptors of the parent compounds T and H; (v) to substantiate that PE, TD, and HD can be considered as new multi-functional components of pharmaceuticals and cosmetics; and (vi) to propose a novel type of meso-carrier for bioactive substances, modeled on the pearl structure, featuring an inorganic core covered with organic coatings that contain active substances within the core and layers.

## 2. Results and Discussion

In order to achieve the primary objectives of this work, the global chemical activity parameters and thermodynamic descriptors of all compounds considered have been calculated. For this purpose, the following quantities defined in the Appendix A are used:(i)Bond dissociation enthalpy BDE, adiabatic ionization potential AIP, proton dissociation enthalpy PDE, proton affinity PA, electron transfer enthalpy ETE, the free Gibbs acidity H_acidity_ (in the gas phase), or G_acidity_ (in hydrophobic, e.g., benzene, and hydrophilic, e.g., water, solvents) [22,23,24,25,26];(ii)The ionization potential IP, electron affinity EA, chemical potential μ, absolute electronegativity χ, molecular hardness η and softness S, electrophilicity index ω, the electro-donating ω^−^ and electro-accepting ω^+^ powers, and the Ra, Rd indexes [27,28,29,30,31,32].

The thermodynamic parameters characterize the radical scavenging potency of the basic compounds and are associated with the following deactivation mechanisms: hydrogen atom transfer (HAT, BDE), single electron transfer followed by proton transfer (SET-PT, AIP, PDE), sequential proton loss electron transfer (SPLET, PA, ETE), and transition metal chelation (TMC, H_acidity_, G_acidity_). They can be calculated by taking advantage of the enthalpies of cation H(R—H^●+^), radical H(R^●^), anion H(R^−^), the parent (neutral) compound H(R—H), and the enthalpies of hydrogen H(H^●^), electron H(e^−^), and proton H(H^+^). A small value of the calculated descriptor indicates a low energy of dehydrogenation (HAT), ionization (SET-PT), and deprotonation (SPLET) in the initial stage of radical deactivation. In the case of two-stage processes, the sum of parameters (PA + ETE or AIP + PDE) should also be taken into account. The geometries of all tested compounds, including their cationic, anionic, and radical forms, were optimized utilizing the procedure delineated in the Materials and Methods section. The optimized geometries of the neutral molecules PE, T, H, TD, and HD are presented in Figure 3.

The values of the thermodynamic descriptors of PE reported in Table 1 reveal that in the gaseous and hydrophobic phases, PE scavenges radicals via the hydrogen atom transfer (HAT) mechanism.

Conversely, in hydrophilic environments, the mechanism shifts to single-electron transfer followed by proton transfer (SETPT). The hydroxymethoxy moiety exhibits significantly greater activity in this context compared to the aldehyde group. This is demonstrated by the parameter values (in [kcal mol^−1^]) PA = 42.25 and their sum PA + ETE = 109.90, which are considerably lower than the corresponding parameters PA = 68.86, and the sum PA + ETE = 120.29 for the aldehyde group. These findings indicate that the hydroxymethoxy moiety in PE is preferentially activated to neutralize free radicals in water. The hydrophobic environment affects the antiradical mechanism preferred by PE, which changes from SPLET to HAT. However, the hydroxymethoxy moiety with BDE = 66.47 [kcal mol^−1^] still shows stronger anti-radical activity than the aldehyde group with BDE = 74.99 [kcal mol^−1^]. The situation changes radically in a vacuum, where the aldehyde group characterized by BDE = 72.00 [kcal mol^−1^] scavenges free radicals by the HAT mechanism, while the hydroxymethoxy moiety of BDE = 100.22 [kcal mol^−1^] is practically inactive in this respect. It is important to emphasize the significant difference in the chelating potential attributed to the two active groups in PE. The aldehyde group characterized by G_acidity_ = 311.86 [kcal mol^−1^] demonstrates minimal participation in the complexation of transition metal cations, in contrast to the hydroxymethoxy group with G_acidity_ = 280.95 [kcal mol^−1^], which exhibits a strong chelating effect in a hydrophilic environment. The above situation also occurs in the hydrophobic medium, in which the hydroxymethoxy moiety with G_acidity_ = 288.37 [kcal mol^−1^] can successfully chelate metal cations. A detailed analysis of the structures of the PE radical and the anion in an aqueous environment (Figure 4) indicates that they adopt the form of a polyendial-formaldehyde complexes.

Studies have demonstrated [3] that polyendials possess two aldehyde groups actively involved in scavenging free radicals. Consequently, PE can neutralize radicals through both a hydroxymethoxy group and the two aldehyde groups, rendering it a highly effective antiradical compound found in nature. The second component of the complex indicates that, during dehydrogenation and deprotonation, PE releases a substance with a broad spectrum of bioactivity and, therefore, practical applications. Formaldehyde (oxymethylene) is utilized as a fumigant in agricultural settings, a hard surface disinfectant in commercial and industrial premises, and a preservative for consumer products (e.g., detergents and cleaners) as well as biological specimens. Additionally, it serves as an antimicrobial agent in vaccines and cosmetics. However, according to Annex V [33], the latter application does not involve free formaldehyde, but instant its releasers, which produce small doses of this antimicrobial agent through hydrolysis. The release process is dependent on the matrix, pH, storage time, and temperature. Annex V [33] allows the following synthetic formaldehyde donors for cosmetic use (number in Annex): hexetidine (19), 5-bromo-5-nitro-1,2-dioxane (20), bronopol (21), imidazolidinyl urea (27), DMDM hydantoin (33), diazolidinyl urea (46), sodium hydroxymethylamino acetate (51), and benzylhemiformal (55). The structure of the PE anion and radical in Figure 4 indicates that this compound can be regarded as a natural formaldehyde donor with the potential application as a multi-functional component in cosmetics, where it may function as a preservative, free radical scavenger, and coloring agent. Given that formaldehyde is released by the HMV moiety, there is a potential for designing a broad class of PE analogues containing this active moiety. The derivatives TD and HD of T and H, depicted in Figure 2, can be considered as examples. T is present in a variety of natural and processed raw materials, for example, in olive oil, in white and red wine [34], in lemons and pepper [35], and in the *Rhodiola* plant species, in which it occurs in the form of glucoside (salidroside) [17,36]. It is characterized by a wide spectrum of biological activities including anti-radical, stress-protective, anti-inflammatory, anti-cancer, cardioprotective, neuroprotective, and many others [36]. H was isolated from the essential oil component of the heartwood of *Taiwanese hinoki* (cypress *Chamaecyparis taiwanensis*). It occurs naturally in the bark and roots of trees from the *Cupressaceae* family, such as the giant cedar (*Thuja plicata*) and Japanese cedar (*Thujopsis dolabrata*). In vitro studies showed that H exhibits anti-radical, anti-inflammatory, and antimicrobial activity [37,38,39]. Because of the ability to chelate polyvalent metal ions, it slows down radical processes and affects the activity of metal-containing enzymes. In vivo, hinokitiol reduces the occurrence of skin photodamage caused by exposure to UV radiation [40]. In modern cosmetology, it is used in skin, hair, and scalp care preparations, anti-discoloration cosmetics, and toothpastes. To assess the activity of the modeled substances, the thermodynamic descriptors of the basic compounds and their modified forms were calculated and are presented in Table 2. Given that the results presented in Table 1 indicate the highest activity for PE in a hydrophilic medium, the calculations for T, H, TD, and HD were performed for water.

The analysis of the obtained results indicates that SPLET is the preferred mechanism for scavenging radicals in an aqueous environment, both for the original compounds and their derivatives. Substituting the ethanol moiety in T with the active group HMV transferred from PE enhanced the free radical scavenging potential of both hydroxyl groups. This assertion is supported by the values of the descriptors PA(TD1) < PA(T1) and PA(TD2) < PA(T2). Furthermore, the total energy input required to activate the first and second stages of SPLET decreased, as demonstrated by the combined value of PA + ETE for both the original T and the modified TD compounds. An intriguing situation arises with the H derivative HD2, for which PA(HD2) < PA(H1) < PA(HD1). This relationship indicates that the attached HMV moiety will first participate in the initial step of the SPLET mechanism. Bearing in mind that ETE(HD2) ≈ BDE(HD2), the activation of the second SPLET stage is energetically competitive with the HAT mechanism. Consequently, in the first stage of SPLET, deprotonation of PE occurs, resulting in the release of formaldehyde, while the second stage of SPLET, or the competitive HAT mechanism, merely deactivates radicals via a hybrid mechanism. It is noteworthy to emphasize the increase in chelating activity of both modified compounds with the transferred HMV moiety.

The values of the global descriptors reported in Table 3 indicate that the polarity of the environment has a significant impact on the increase in PE activity. It is reflected in the decrease in the ∆E = E(LUMO) − E(HOMO) value from 2.0422 [eV] to 1.9121 [eV] when transitioning from the gas to the water phase. A small HOMO-LUMO energy gap characterizes a soft molecule that is less stable and more reactive. In the case of EA and IP parameters, a smaller activating effect of the hydrophilic environment, associated with a decrease in the IP value from 5.0662 to 4.9653 [eV] and an increase in EA from 3.0240 to 3.0531 [eV], is observed. The small IP value indicates a greater tendency of the molecule to participate in the chemical reaction related to electron transfer, whereas large values of EA characterize a greater capacity of a compound to accept electrons and to convert it into the anionic form. The descriptors characterizing the electro-donating ω^-^ and electro-accepting ω^+^ power of PE increase from ω^−^ = 10.1625 up to 10.5302 [eV] and ω^+^ = 6.1174 up to 6.5210 [eV] in the hydrophilic medium. This indicates that the antioxidant activity of PE diminishes in water, in contradistinction to their anti-reductant activity, which increases. These conclusions are consistent with the acceptance Ra and donation Rd indexes, which take the values Ra = 1.7982 (gas) and 1.9168 (water), and Rd = 2.9288 (gas) and 3.0348 (water). Consequently, PE in all media considered is a more effective electron acceptor than F (Ra = 1) and a more effective electron donor than Na (Rd = 1). It is also worth noting that the activity of PE is greater than that attributed to astaxanthin [5]: Ra = 0.94, Rd = 2.10, ω^+^ = 3.21, ω^−^ = 7.27, which is recognized as the most effective electron acceptor among the pigments identified in nature [5]. The electrophilicity index ω takes the values from 8.0123 [eV] in the gas phase up to 8.4061 [eV] in the water medium. Because this parameter can be related to the toxicity expressed, for example, in the lethal dose LD_50_ of the compound [41], one may expect a very low toxicity of PE in comparison to the highly toxic HCN (LD_50_ = 4.11 [mg kg^−1^], ω = 0.0982 [eV]) and its low toxic salt CuCN (LD_50_ = 1265.0 [mg kg^−1^], ω = 0.26991 [eV]). This fact is promising for the potential application of PE as an ingredient in food products and supplements. The analysis of the remaining parameters of the chemical activity (χ, η, S) reveals their weak dependence on the medium type: in water, χ and η decrease, whereas S increases their values. This proves that the medium type affects the hardness S of the molecule, i.e., susceptibility to deformation or polarization of the electron cloud under the influence of external factors (reagents), and leads to a decrease. The results collected in Table 3 enable the analysis of the influence of the type of functional on the values of PE chemical activity descriptors. The differences in parameter values are significant, but they do not have an impact on the interpretation of the PE activity. The exception is the Ra parameter, which for the M06-2X and BHandHLYP functionals takes the values less than one, which predicts that PE is a less effective electron donor than Na (Rd = 1).

Analysis of the chemical activity parameters from Table 4 provides important information on both the known T and H compounds and their not yet synthesized TD and HD derivatives. First of all, attaching the HMV moiety to the H tropolone ring only slightly influenced the parameter values and, consequently, the HD activity. This means that the natural antiradical properties of H will be maintained in HD at the current level, while the antimicrobial activity is expected to be enhanced by an additional microbicidal factor that can be attributed to the released formaldehyde. In the case of T, we observe a significant impact of the structure modification on the TD parameter values. In particular, ∆E = E(LUMO) − E(HOMO) decreases from 5.6981 [eV] to 4.6393 [eV], and the IP value changes from 6.3022 to 5.5936 [eV], whereas the value of EA increases from 0.6033 to 0.9543 [eV]. This indicates a greater tendency of TD to electron transfer and a greater capacity of this compound to accept of electrons. The electro-donating ω^-^ and electro-accepting ω^+^ power of TD increase with respect to T from ω^−^ = 4.1744 up to 4.2373 [eV] and ω^+^ = 0.7217 up to 0.9634 [eV]. Consequently, the antioxidant activity of TD diminishes, whereas the anti-reductant activity increases with respect to T. This tendency is confirmed by the Ra and Rd indexes, which take the values Ra = 0.2121 (T) and Ra = 0.2832 (TD), and Rd = 1.2031 (T) and Rd = 1.2212 (TD). Thus, T and TD are less effective electron acceptors than F (Ra = 1) and more effective electron donors than Na (Rd = 1). A further analysis reveals that χ and η decrease, whereas S increases their following the modification of T to TD. Hence, the hardness S of the TD is greater than the parent compound T, in contradistinction to softness η and the electronegativity χ, which decrease.

## 3. Materials and Methods

Following the fact that PE possesses a chemical structure analogous to pisttacofulvin (polyenal responsible for the parrot feather coloration [3]) characterized by ten conjugated double bonds and a terminal methyl group substituted by the hydroxymethoxy moiety, the calculations were performed at the identical DFT/B3LYP/QZVP level of theory and C-PCM solvation model as in the study [3]. This approach utilizes Becke’s [42] exchange functional in conjunction with the Lee–Yang–Parr [43] one, the Weigend–Ahlrichs valence quadruple-zeta polarization basis set (QZVP) [44], and the Conductor-like Polarizable Continuum Model (C-PCM) [45]. The test calculations performed with other functionals [46,47] provided the following total energy values of PE: E(M06-2X) = −1078.364604 [Ha] and E(BHandHLYP) = −1078.160058 [Ha] in comparison to E(B3LYP) = −1078.854040 [Ha]. The difference in energy values is E(M06-2X) − E(B3LYP) = 307.13 [kcal mol^−1^], and E(BHandHLYP) − E(B3LYP) = 435.48 [kcal mol^−1^], respectively, which proves that the B3LYP functional in combination with the QZVP basis set and the C-PCM model is optimal for the PE compound under consideration. The same level of theory was also applied to the remaining compounds (T, H, TD, HD) to allow a reliable comparison of their activities. To estimate the effect of functional type on the values of the chemical activity parameters, the calculations were also performed using the M06-2X and BHandHLYP functionals, commonly used in the determination of activity descriptors. The calculations were performed for PE in the gas phase as well as in the hydrophobic (benzene) and the hydrophilic (water) media. The choice of benzene was dictated by the very small ε = 2.2706 value of the dielectric constant (relative permittivity), which reflects well the hydrophobic nature of this solvent. Test calculations showed that the use of other lipophilic solvents with similar ε-values (cyclohexane ε = 2.0165, heptane ε = 1.9113, 2,4-dimethylpentane ε = 1.8939) had no effect on the values of the evaluated parameters. To calculate enthalpies H(R—H^●+^), H(R^●^), H(R^−^), H(R—H), and the HOMO-LUMO energies, indispensable in the determination of thermodynamic and chemical activity descriptors, we employed the DFT method implemented in Gaussian vs 16 software application [48]. The input structures were constructed by taking advantage of the Gauss View-6.1 graphical interface [48], whereas the calculations were carried out in the Supercomputing and Networking Center of Poznań. Since the computations for the compounds studied at the B3LYP/QZVP level are time-consuming and their convergence depends on the initial geometry input, we used a strategy similar to that in our previous work [3]. In this approach, the optimization was divided into three stages: (i) the determination of an approximate geometry at the B3LYP/6311++G(d,p) level of the theory in the gas phase; (ii) the geometry refinement at the B3LYP/cc-pVQZ level in the gas phase; and (iii) the final calculation at the B3LYP/QZVP level in the gas phase or in the water (benzene) medium. Thus, the optimization was performed at each stage, with the geometry determined at the lower level serving as the starting point for optimization at the higher level. The enthalpies H(R—H^●+^), H(R^●^), H(R^−^), and H(R—H) and the HOMO-LUMO energies generated in this scheme were used in the calculation of the activity descriptors defined in the Appendix A. The final values of the descriptors were calculated using the Maple vs 16 processor for symbolic computations. In the calculations, we used the following values (in Hartree [Ha] unit) of the electron, proton, and hydrogen enthalpies in the gas phase [32]: H(e^−^) = 0.001198, H(H^+^) = 0.002363, H(H^●^) = −0.497640; water: H(e^−^)_aq_ = −0.03879545, H(H^+^)_aq_ = −0.38690958, H(H^●^)_aq_ = −0.49916356; benzene: H(e^−^)_be_ = −0.00146823, H(H^+^)_be_ = −0.33815566, H(H^●^)_be_ = −0.495202304. The last six values can be calculated using the following relationships:H(e^−^)_sol_ = H(e^−^) + ∆_sol_H(e^−^), H(H^+^)_sol_ = H(H^+^) + ∆_sol_H(H^+^), H(H^●^)_sol_ = H(H^●^) + ∆_sol_H(H^●^)
in which ([kJ mol^−1^] unit) ∆_aq_H(e^−^) = –105, ∆_aq_H(H^+^) = –1022, ∆_aq_H(H^●^) = –4.0, ∆_be_H(e^−^) = –7, and ∆_be_H(H^+^) = –894, ∆_be_H(H^●^) = 6.4 are the solvation corrections recommended by Rimarčik et al. [22].

## 4. Conclusions

The calculations indicate that in a hydrophilic environment, PE scavenges radicals through the SPLET mechanism, whereas in a hydrophobic medium, it operates via the HAT mechanism. The hydroxymethoxy moiety is more active in this process than the aldehyde group. During its deprotonation (dehydrogenation), an unstable anion (radical) is formed, which decomposes with the release of formaldehyde. This compound exhibits strong antimicrobial and preservative properties, demonstrating that PE functions in pearls not only as a dye but also as a protector against microorganisms and free radicals. Through evolution, mollusks have developed an effective defense mechanism that shields their tissues from parasites and damage by organic and inorganic elements, forming a protective barrier around an irritant in the form of a pearl. As argued by Shi et al. [15], PE is incorporated in the layers of conchiolin protein and is not located within the crystal lattice of aragonite (calcite), the primary structural components of pearls. Therefore, the pearl can be considered a carrier of the active PE compound, which neutralizes harmful free radicals and releases microbicidal formaldehyde, thereby preserving bio-irritants and protecting the mollusk from microorganisms. Consequently, a macroscopic pearl can serve as a model for a new class of mesoscopic carriers that are a combination of spherulities^TM^ (multilayer delivery system [49]) and hydroxysomes^TM^ (nanoporous calcium phosphate delivery system [50]). Meso-carriers, modeled structurally on pearls and comprising a multilayer component with PE, TD, or HD as formaldehyde releasers embedded within the layers, along with other substances such as anti-irritants and fragrances within the mineral component, appear to be an optimal multi-functional ingredient for cosmetics, particularly deodorants. One issue requiring clarification is the occurrence of white pearls that do not contain PE. If the proposed interpretation of PE’s function in pearls is correct, then the absence of PE in these pearls may be attributable to an albino genetic mutation or the nature of the irritant. In the latter case, only bio-irritants (e.g., parasites) would necessitate the presence of PE in pearls. This intriguing issue warrants further research, and the explanation proposed requires experimental verification. The calculated chemical and thermodynamic activity descriptors indicate that the natural basic compounds T and H, as well as their projected derivatives TD and HD, are anti-reductants, scavenging free radicals via the SPLET mechanism in hydrophilic environment—a natural medium for their bioactivity. The antiradical activity of T can be enhanced, whereas the antimicrobial activity of H can be amplified by modifying their structures through the transfer of the HMV moiety from PE to T and H. The new compounds modeled in this way can also be applied as multi-functional components of cosmetics, potentially replacing current formaldehyde releasers, whose harmfulness and permissible concentrations are the subject of intense discussion [51].

## Figures and Tables

**Figure 1 molecules-29-03414-f001:**
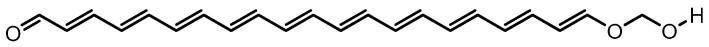
Polyenal 21-(hydroxymethoxy)henicosadecaenal identified in the freshwater pearls from mollusk *Hyriopsis cumingii* by Shi et al. [15].

**Figure 2 molecules-29-03414-f002:**
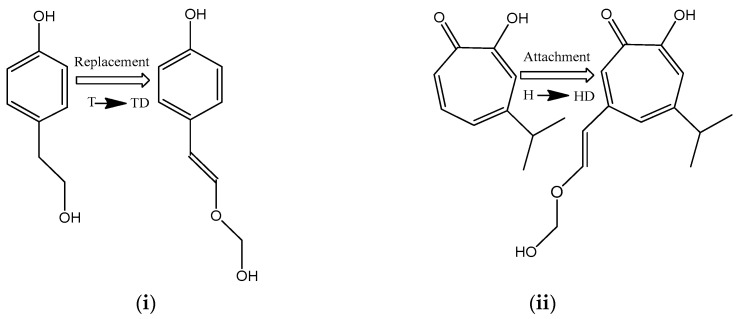
Tyrosol (T) and hinokitiol (H) derivatives 4-(2-(hydroxymethoxy)vinyl)phenol (TD), 6-(2-(hydroxymethoxy)vinyl)-4-isopropyl tropolone (HD) generated by (**i**) the replacement of 2-hydroxyethyl group in T by the 2-(hydroxymethoxy)vinyl (HMV) moiety, and (**ii**) the attachment of HMV to the tropolone ring in H.

**Figure 3 molecules-29-03414-f003:**
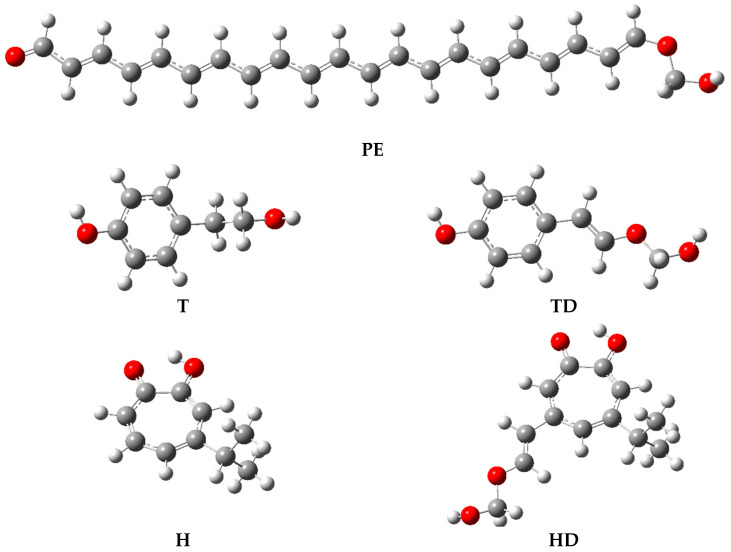
The optimized geometries of PE identified in pearls from the mollusk *Hyriopsis cumingii*, and derivatives (TD, HD) of tyrosol (T) and hinokitiol (H) in the water medium evaluated by the DFT method at the B3LYP/QZVP theory level, using C-PCM solvation model.

**Figure 4 molecules-29-03414-f004:**
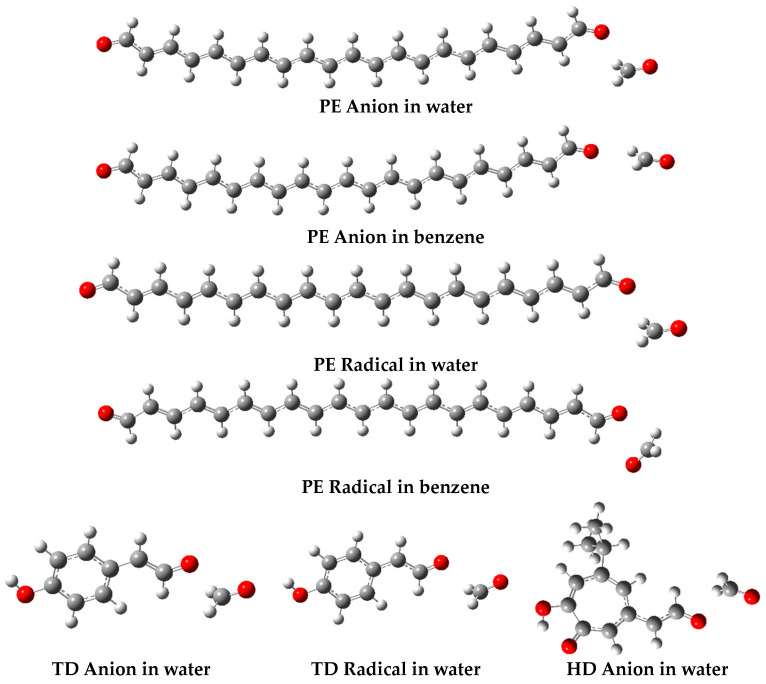
The optimized geometries of the cations and radicals of PE and derivatives (TD, HD) of tyrosol (T) and hinokitiol (H) in the hydrophilic (water) and hydrophobic (benzene) media evaluated at the DFT/B3LYP/QZVP theory level, using C-PCM solvation model. The HD radical in water is stable and does not form the complex with formaldehyde.

**Table 1 molecules-29-03414-t001:** The thermodynamic descriptors [kcal mol^−1^] of PE in the vacuum, benzene and water medium evaluated at the B3LYP/QZVP theory level, using C-PCM solvation model. Two active moieties O=C–H and C–O–H in PE have been taken into account.

Descriptor	O=C–H			C–O–H		
	Vacuum	Benzene	Water	Vacuum	Benzene	Water
BDE	72.00	74.99	74.19	100.22	66.47	63.80
PA	337.92	110.43	68.86	308.24	82.53	42.25
ETE	48.59	62.19	51.43	106.49	81.57	67.65
PA + ETE	386.51	172.62	120.29	414.73	164.10	109.90
AIP	136.74	119.46	83.24	136.74	119.46	83.24
PDE	249.76	53.16	37.04	277.99	44.63	26.65
G_acidity_	–	322.90	311.86	–	288.37	280.95
H_acidity_	336.44	–	–	306.76	–	–
Mechanism	HAT	HAT	SPLET	HAT	HAT	SPLET

**Table 2 molecules-29-03414-t002:** The thermodynamic descriptors [kcal mol^−1^] characterizing tyrosol T1, T2, and hinokitiol H1 derivatives TD1, TD2, HD1, HD2, determined at the DFT/B3LYP/QZVP level of the theory in the water, using the C-PCM solvation model. The numbers 1 and 2 describe the hydroxyl group attached (1) to the ring, and (2) to the aliphatic carbon.

Descriptor	T1	T2	TD1	TD2	H1	HD1	HD2
BDE	81.86	101.06	74.83	74.87	87.18	87.73	100.07
PA	52.36	68.63	51.08	53.28	47.38	49.50	45.48
ETE	75.60	78.53	69.85	67.68	85.89	84.32	100.68
PA + ETE	127.96	147.16	120.92	120.97	133.28	133.82	146.16
AIP	115.18	115.18	96.66	96.66	118.14	114.22	114.22
PDE	12.78	31.98	4.26	24.30	15.13	19.61	31.94
G_acidity_	295.30	312.20	294.49	294.05	290.20	291.36	285.27
Mechanism	SPLET	SPLET	SPLET	SPLET	SPLET	SPLET	SPLET

**Table 3 molecules-29-03414-t003:** The global chemical activity descriptors [eV] of PE determined in the vacuum, benzene, and water medium at the DFT/B3LYP, M062-X, BHandHLYP/QZVP theory levels, using C-PCM solvation model.

Functional	B3LYP	B3LYP	B3LYP	M06-2X	BHandHLYP
Descriptor	Vacuum	Benzene	Water	Water	Water
EA	3.0240	3.0297	3.0531	2.1590	1.8634
IP	5.0662	5.0189	4.9653	4.9653	5.7574
ΔE	2.0422	1.9892	1.9121	2.8063	3.8939
η	1.0211	0.9948	0.9561	1.4032	1.9470
S	0.4897	0.5027	0.5230	0.3563	0.2568
χ = −μ	4.0451	4.0243	4.0092	3.5621	3.8104
ω	8.0123	8.1416	8.4061	4.5215	3.7287
ω^+^	6.1174	6.2538	6.5210	2.9158	2.0668
ω^−^	10.1625	10.2781	10.5302	6.4779	5.8772
Ra ^[a]^	1.7982	1.8383	1.9168	0.8571	0.6075
Rd ^[a]^	2.9288	2.9621	3.0348	1.8669	1.6938

[a] Dimensionless parameter.

**Table 4 molecules-29-03414-t004:** The global chemical activity descriptors [eV] of tyrosol (T), hinokitiol (H), and their derivatives TD and HD determined in the water medium at the DFT/B3LYP/QZVP theory level, using C-PCM solvation model.

Descriptor	T	TD	H	HD
EA	0.6033	0.9543	2.1508	2.1617
IP	6.3022	5.5936	6.4415	6.3329
ΔE	5.6989	4.6393	4.2907	4.1712
η	2.8494	2.3196	2.1453	2.0856
S	0.1755	0.2156	0.2331	0.2397
χ = −μ	3.4527	3.2739	4.2961	4.2473
ω	2.0919	2.3104	4.3016	4.3247
ω^+^	0.7217	0.9634	2.4217	2.4618
ω^−^	4.1744	4.2373	6.7178	6.7091
Ra ^[a]^	0.2121	0.2832	0.7118	0.7236
Rd ^[a]^	1.2031	1.2212	1.9361	1.9336

[a] Dimensionless parameter.

## Data Availability

All new data created are reported in this work.

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
