# Peer review of "Enhancing Bioactivity through the Transfer of the 2-(Hydroxymethoxy)Vinyl Moiety: Application in the Modification of Tyrosol and Hinokitiol"

_molecules, 2024, doi:10.3390/molecules29143414_

Round 1

Reviewer 1 Report

Comments and Suggestions for Authors

Dear author and editor,

I had the opportunity to read the manuscript entitled Enhancing bioactivity by transferring active moiety, submitted for publication in Molecules.

For the first moment, I was attracted by the title, it is very suggestive, but, after reading the all manuscript, I recommend to include in the title some words related, either to the antiradical and antimicrobial properties, either to the active moity that is subject to these studies.

The study case in this paper constitutes compounds containing O=C‒H and C‒O‒H, susceptible to release formaldehyde, that exhibit antimicrobial properties, namely Tyrosol (T) and hinokitiol (H) derivatives. A analysis of the global reactivity parameters, according Koopman’s theorem and thermodynamics properties are carried on, from computed data obtained using B3LYP/6311++G(d, p) level of the theory using C-PCM solvation  model, in the gas phase, in water and in benzene.

Koopmans' theorem must be given as reference for the calculus of quantum chemical reactivity parameters starting from energies of frontier molecular orbitals.

Author should explain the choice of benzene as media for computation properties, and why not other solvent to mimic the hydrophobic medium.

I consider the manuscript for publication with the above-mentioned minor revisions.

Reviewer 2 Report

Comments and Suggestions for Authors

The submitted manuscript is a purely theoretical work, which is OK, presenting the results of the DFT calculations (quite routine ones) for the 5 medium size organic compounds. The calculated properties (HOMO-LUMO, hardness, etc.) are quite routine and not compared with the experimental results, not even with the literature ones. Therefore, the works require major revisions.

Major issues:

The title is too general, it doesn’t provide any information about the object of the study nor the methods applied.

Number of references (60!) is too large and should be greatly reduced.

The introduction is very strange. First, we’ve got (too) general part about the products with biological activity. Then, most of the introduction is about a compound avvreviated PE (polyenal 21-(hydroxymethoxy)henicosadecaenal). And when the reader thinks that this work will be about this compound, in the final part of the review there is a sudden information “hey, in this work we won’t model the PE modifications but other compounds with barely no connection to the PE, apart from the possession of CHO and OH groups, shared but many other compounds”. This lacks any sense.

Line 54, work [25] is over 10 years old, how can this compound be called “new”?

Line 83, I don’t agree, actually calcite not aragonite is present in most of the shells.

Lines 80-81 and 87-88, the same phrases are being repeated. This should be rewritten.

The aim of the study is not clearly stated. It lacks a sentence like “The aim of this study was to….”

Lines 152-153, I don’t know why the Authors haven’t calculated the vibrational frequencies to obtain the ZPVE and Gibbs free energy?

Line 157, “extremely” is not an appropriate adjective. It would be if the Authors had used CC, MP2, G4, or any other methods at higher level of theory than DFT.

Lines 157-158, “and their convergence depends on the initial geometry input” – this is valid for all QC methods, not only this one…

Figure 3, why the Authors haven’t studied any derivatives of PE?

Lines 368-370: “This compound exhibits strong antimicrobial and preservative properties, demonstrating that PE functions in pearls not only as a dye but also as a protector against microorganisms and free radicals.” – those are definitely NOT the conclusions from this work as the Authors haven’t conducted such studies…

Minor comment:

Line 102, it should be H and HD

Round 2

Reviewer 2 Report

Comments and Suggestions for Authors

The Author has answered sufficiently on most of my comments. From the technical point of view, the work is acceptable.